# Conformational plasticity of NaK2K and TREK2 potassium channel selectivity filters

Marcos Matamoros[1,2], Xue Wen Ng[1,2], Joshua B. Brettmann[1,3], David W. Piston ®[1,2] & Colin G. Nichols ®[1,2] ✉

The K[+] channel selectivity filter (SF) is defined by TxGYG amino acid sequences that generate four identical K[+] binding sites (S1-S4). Only two sites (S3, S4) are present in the non-selective bacterial NaK channel, but a four-site K[+]-selective SF is obtained by mutating the wild-type TVGDGN SF sequence to a canonical K[+] channel TVGYGD sequence (NaK2K mutant). Using single molecule FRET (smFRET), we show that the SF of NaK2K, but not of non-selective NaK, is ion-dependent, with the constricted SF configuration stabilized in high K[+] conditions. Patch-clamp electrophysiology and non-canonical fluorescent amino acid incorporation show that NaK2K selectivity is reduced by crosslinking to limit SF conformational movement. Finally, the eukaryotic K[+] channel TREK2 SF exhibits essentially identical smFRET-reported ion-dependent conformations as in prokaryotic K[+] channels. Our results establish the generality of K[+]-induced SF conformational stability across the K[+] channel superfamily, and introduce an approach to study manipulation of channel selectivity.

Differential concentrations of potassium (K[+]) and sodium (Na[+]) ions across cell membranes are crucial to generating the chemical and electrical activity that underlies life itself, and unmasking the basis for exquisite differentiation between these two similar ions at the cell membrane has long been a major focus of biophysicists and structural biologists[1]. The first ion channel crystal structure, from the highly K[+] selective bacterial KcsA channel, showed that the narrowest part of the pore, formed by the backbone carbonyls of the TxGYG channel signature sequence, generates four identical potassium binding sites, S1-S4 in the K[+] channel selectivity filter (SF)[2–4]. A constellation of K[+] channel structures have followed, including Inwardly rectifying K[+] channels[5–7], Two-Pore-Domain K[+] channels (K2P)[8–10], Voltage gated K[+] channels[11–14] and Ligand-gated K[+] channels[15,16], all revealing a nearly identical four K[+] binding site structure of the SF. Other ion channels, such as the hyperpolarization-activated cyclic nucleotide-gated (HCN) channels, permit a significant Na[+] permeability, yet have very similar SF sequences. In HCN1, the filter adopts a non-canonical configuration where only S3 and S4 cation-binding sites are formed[17]. This S3-S4 configuration is also seen in the prokaryotic NaK channel, in which the SF sequence TVGDGN generates a non-selective channel[18].

Significantly, the canonical K[+] selective four ion binding sites configuration at the SF can be obtained by mutating the NaK SF sequence to TVGYGD (NaK2K mutant), generating a fully K[+]-selective channel.

While structural analyses using crystallography, and now Cryo-EM techniques, provide unprecedentedly detailed views of channel structures[19,20], they provide essentially no kinetic information, potentially leading to underappreciation of any structural flexibility underlying selectivity and permeation. To obtain a full understanding of living ion channel function, it is essential for channel structure-function relationships to be dynamically investigated in functioning channels within lipid membranes. Using single-molecule FRET (smFRET) to do this, we have previously demonstrated that the SF of the bacterial KirBac1.1 K[+] channel transits between constrained and dilated conformations as a function of the ionic milieu[21,22]. The constrained K[+]-selective conformation is induced by K[+] ions themselves; in their absence the SF adopts wider and less stable conformations. As well as potentially explaining the lack of K[+]-channel SF crystal structures in K[+]-free conditions[23–26], these results suggest a K[+]-induced conformational basis of selectivity in all K[+] channels. To date, however, such experiments have been limited to the

[1]Center for Investigation of Membrane Excitability Diseases, Washington University School of Medicine, St. Louis, MO, USA. [2]Department of Cell Biology and Physiology, Washington University School of Medicine, St. Louis, MO, USA. [3]Present address: Millipore-Sigma Inc., St. Louis, MO, USA.
✉e-mail: cnichols@wustl.edu

experimentally tractable KirBac1.1 model[21,22], and require confirmation in other systems.

In the present study, we labeled several well characterized cation channels, the prokaryotic NaK and NaK2K[27,28] and eukaryotic TREK2 (Two-Pore-Domain K+ channel (K2P) TREK2)[10,29,30], at the SF and then examined structural dynamics. Our results indicate that the SF of K+ selective NaK2K and TREK2 (K+ selective), but not that of non-selective NaK, also shifts from constricted to dilated and dynamic SF conformations in Na+ versus K+ conditions, implying similar dynamics for the K+ channel SF, from bacteria to humans. In addition, molecular crosslinking experiments show that ion selectivity of NaK2K can be changed by restricting the conformational flexibility of the SF, illustrating a potential avenue for future manipulation of channel selectivity.

## Results

### K+-selective NaK2K, but not non-selective NaK, shows ion-dependence of SF conformation

One of the most important models in the study of ion selectivity has been the NaK (Na+- and K+-conducting) channel from *Bacillus cereus*, a non-selective tetrameric cation channel that shares high homology and a similar structure with the bacterial KcsA K+ channel, but whose SF adopts a different architecture. NaK crystal structures show that, unlike the K+ channel SF signature sequence (TxGYG), which contains four K+-binding sites (S1-S4), the NaK channel SF (TVGDGN) only generates two cation-binding sites (equivalent to sites S3 and S4, Supplementary Fig. 1A)[18], but a highly selective, four K+ ion binding site configuration can be obtained by mutating the NaK SF sequence TVGDGN to a canonical K+ channel SF signature sequence TVGYGD (NaK2K mutant, Supplementary Fig. 1A)[24,27,28]. We engineered, expressed, labeled, and reconstituted tandem dimeric NaK and NaK2K constructs, each dimer containing only 1 cysteine (at position T73), resulting in 2 diagonally opposed T73C residues within the tetrameric channel (Fig. 1A, see Methods). Tetramer fractions of Alexa Fluor 555 and Alexa Fluor 647 c2 maleimide-labeled channels were collected by size exclusion chromatography and reconstituted into liposomes (POPE:POPG = 3:1). Fluorescence monitored ion flux assays confirm that all labeled mutants retain channel activity (Supplementary Fig. 1B).

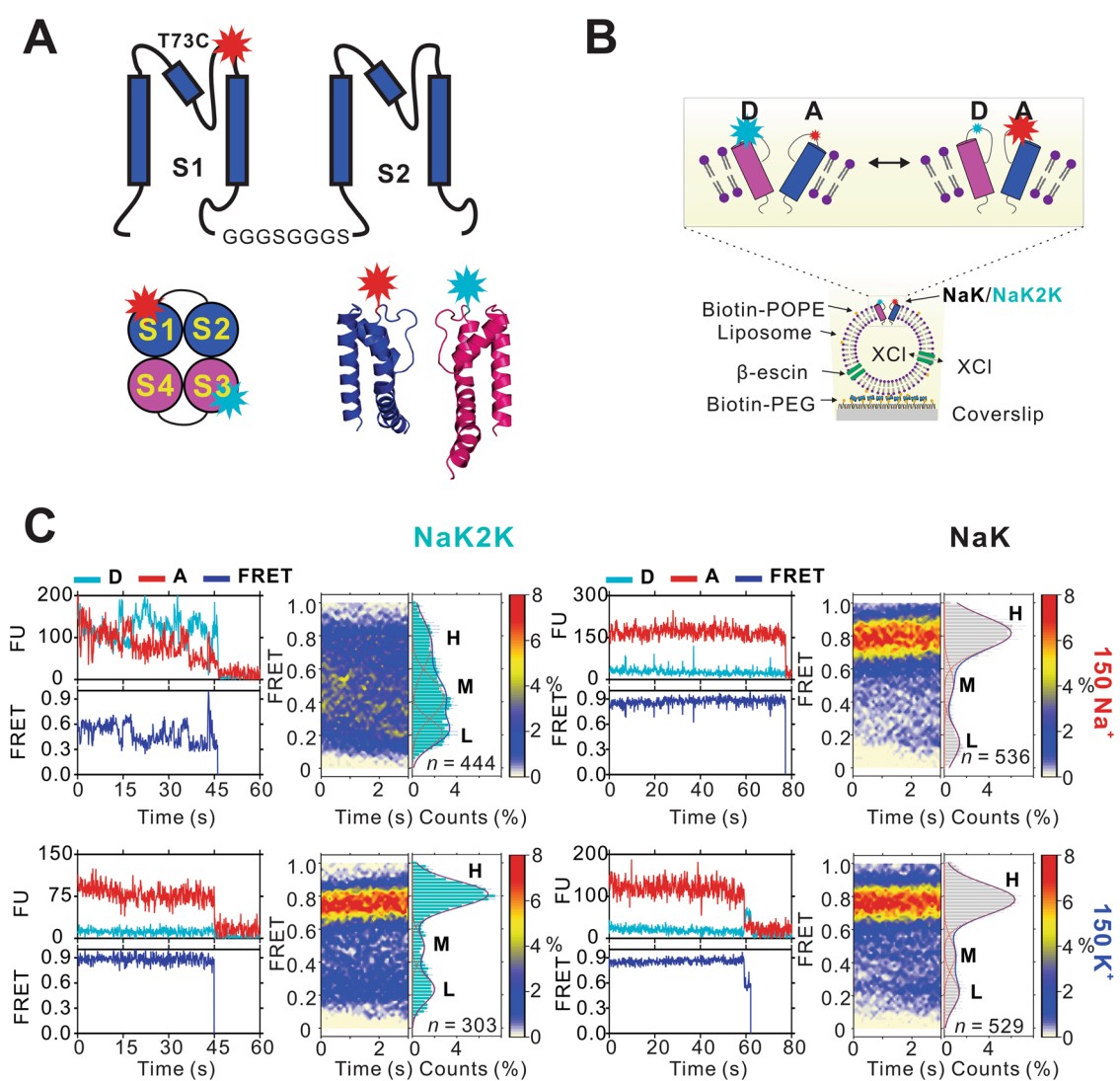

**Fig. 1 | NaK and NaK2K selectivity filter dynamics. A** Strategy for introducing two free cysteines within the tetrameric NaK channel. Organization of the tandem dimeric construct where the cysteine is introduced at position T73 at the top part of the first subunit (S1) TM2 domain (PBD 3E8H). Every linker contains two GGGS amino acid repeats. **B** Single-molecule FRET imaging of NaK proteins in liposomes labeled with Alexa Fluor 555/647 c2 maleimide pair. Proteoliposomes were immobilized on a PEG-coated coverslip surface with biotin-POPE, and then permeabilized by 50 µM β-escin to achieve symmetrical ionic conditions. **C** Representative donor (blue), acceptor (red) and smFRET (dark blue) traces for NaK T73C and NaK2K T73C, as well as FRET contour plots and histograms (*n* = 536 and 529 NaK traces; 444 and 303 NaK2K traces, in the presence of 150 mM NaCl or KCl, respectively).

Using smFRET, we assessed single membrane protein conformational changes in real time within synthetic lipid membranes. smFRET imaging was performed on single molecules incorporated into liposomes and immobilized on the stage of a TIRF microscope (Fig. 1B, see Methods). The potassium selective mutant, NaK2K, shows a very clear ion dependence of the SF, with a constrained stable high FRET signal predominating in 150 mM K[+], but less constrained dynamic low FRET signals in 150 mM Na[+], very similar to those seen in KirBac1.1 labeled at the same location (Fig. 1C)[21,22]. There is no such ion dependence in NaK, indicating an apparent ion independence of the SF conformations. Interestingly, in contrast to our previous findings in SF-mutated non-selective KirBac1.1 mutants, NaK shows predominantly constrained and stable high FRET signals in both 150 mM K[+] and 150 Na[+] (Fig. 1C). The FRET amplitude distributions of both NaK and NaK2K SF (Fig. 1C) show clear peaks, requiring a minimum sum of three Gaussians for adequate fitting, in both 150 mM K[+] or Na[+], with FRET peaks around 0.2, 0.45, and 0.8 (Fig. 1C). If the trace data are very dynamic, cross-correlation analysis between donor and acceptor fluorescence intensities can infer dynamic flexibility between sites to which the specific fluorophore pair is attached[31]. NaK2K exhibited a strong donor-acceptor cross-correlation with a decay time constant of 1.22 s in 150 mM Na[+] but essentially no cross-correlation in 150 mM K[+] (Supplementary Fig. 1C). This indicates marked flexibility of the outer pore of NaK2K in 150 mM Na[+], whereas lack of cross-correlation for NaK in either condition indicates a relative lack of flexibility (Supplementary Fig. 1C).

While inevitably an oversimplification, the FRET amplitude distributions are generally well fit with the sum of three Gaussians. With the caveat that changes in the FRET signal could arise from changes in anisotropy rather than separation of the fluorophores, the observed smFRET amplitude distributions (Fig. 1C) are consistent with the diagonally apposed subunits that contribute to the FRET signal being predominantly dilated, i.e., the fluorophores are relatively distant from the pore axis (generating the low FRET 0.2 state, L, or the intermediate FRET 0.45 state, M), or are closer to the pore axis (generating the high FRET 0.8 state, H). We analyzed the kinetics of idealized concatenated FRET records (Fig. 2, see Methods) from randomly concatenated trajectories[32]. Consistent with K[+] ions stabilizing the constrained SF configuration, the average H state lifetimes for NaK2K, were significantly higher in 150 mM K[+] than in 150 mM Na[+] (Fig. 2B). In contrast, state lifetimes were independent of ionic milieu in NaK (Fig. 2D).

## K[+]-selective NaK2K SF, but not the non-selective NaK SF, is conformationally dynamic in living cell membrane

The above smFRET results show that, just as in KirBac1.1, ion dependence of conformational dynamics is only present in the highly K[+] selective NaK2K, and is absent from the non-selective NaK. The smFRET technique is limited practically to synthetic systems and cannot be carried out in living cell membranes, obviating simultaneous or parallel electrophysiological experiments to confirm the correlation of structural conformational measurements with ion conductance measurements in identical conditions. To circumvent this, we developed non-canonical amino acid (L-Anap) incorporation to monitor SF conformations in NaK and NaK2K in mammalian cell lines. The small size of L-Anap, its environment-sensitive fluorescence (L-Anap shifts its emission spectra to higher wavelengths when the environment becomes more hydrophilic), and the ability to introduce it at specific sites in proteins using an orthogonal amber suppressor tRNA/aminoacyl-tRNA synthetase (aaRS) pair[33], facilitates assessment of bulk ion channel conformational dynamics in membrane proteins[34–37] and in mammalian cell lines[33]. Delivery of functional bacterial membrane proteins to the plasma membrane of mammalian cell lines can be difficult[38], but by introducing a membrane exporting signal (FCYENEV)[39,40] in the C-terminus of NaK (Fig. 3A and Supplementary Fig. 2), eGFP-tagged NaK and NaK2K constructs were functionally

expressed in the plasma membrane of COSm6 cells (Supplementary Fig. 2). To improve the NaK L-Anap protein fluorescence signal-to-noise ratio, we generated giant plasma membrane vesicles (GPMVs)[41,42] (Fig. 3A and Supplementary Fig. 2B). L-Anap was inserted at positions V45 (Transmembrane domain 1, TM1), A53 (Pore Helix, PH) and G76 (Transmembrane domain 2, TM2) in NaK and NaK2K expressed in CosM6 cells (Fig. 3A). These sites were selected to meet two requirements: (1) to be at the external part of the SF and (2) to be at the interphase between the lipid membrane and the aqueous media, to capture any ion-dependent conformational shifts (see Methods). In NaK2K, the emission peak at both V45 and G76 (but not A53) positions were both significantly shifted when 150 mM Na[+] in the extracellular solution was replaced with 150 mM K[+] (Fig. 3B, C). Left-shift for V45-labeled and right shift for G76-labeled suggests these residues move to more hydrophobic and more hydrophilic local environments, respectively, in K[+], potentially reflecting an outward movement of the M2 helix relative to M1. Significantly, no ion dependent shifts were detected at any of these three positions in NaK.

Conceivably, L-Anap incorporation could itself affect channel selectivity and functionality. However, fluorescent imaging using the voltage sensor DiBac showed that both V45 and G76 L-Anap-labeled channels were functional and maintained expected ion selectivity, with WT, and V45- or G76-labeled NaK2K causing similar K[+]-conductance driven hyperpolarization on exposure to low [K[+]], while there was no hyperpolarization of labeled or unlabeled NaK-expressing cells (Supplementary Fig. 2C). This was not the case for L-Anap labeled A53, with both NaK and NaK2K mutants causing marked depolarization, suggesting a loss of K[+]-selectivity, and potentially explaining lack of K[+] dependence of fluorescence for NaK2K labeled at this position.

## Manipulating NaK2K selectivity

Assuming isotropy, with an orientation factor $\kappa^2 = 2/3$, FRET efficiencies of ~0.8, ~0.45 and ~0.2 predict distances of 40, 52 and 64 Å between fluorophore centers, for the H, M and L states, respectively. Although NaK2K has been crystallized in different conditions, the predicted T73 Cα-Cα diagonal distances are ~35 Å in high K[+] or in high Na[+] [27,43], similar to the high FRET efficiency predicted distances. Medium and low FRET states could then represent occasional SF expansions in NaK2K channels, potentially to less or non-selective states. We attempted to trap the SF in an expanded state by crosslinking experiments in NaK2K T73C mutant channels (formed as tetramers of monomeric, not tandem dimeric, constructs, see Supplementary Fig. 3A) using a PEG compound with maleidimide groups attached to both ends of ~6 nm length, close to the low FRET predicted distances (Fig. 4A). Patch-clamp recordings from asolectin liposomes show a significant decrease in single-channel conductances in K[+], and increase in Na[+], when the compound is crosslinked to T73C residues in the external part of the NaK2K selectivity filter, making the channel less K[+] selective and more prone to conduct Na[+] ions (Fig. 4B–D) but not changing NaK ion conduction (Fig. 4D and Supplementary Fig. 3C). Furthermore, another PEG compound with maleidimide groups attached to both ends of ~3 nm length, close to the high FRET predicted distances (Supplementary Fig. 3B), failed to alter NaK or NaK2K selectivity (Supplementary Fig. 3D), even though both the 6 nm and 3 nm PEG compounds successfully crosslink the SF of NaK and NaK2K (Supplementary Fig. 3E). While more extensive studies will be needed to fully understand the partial loss of selectivity induced by the 6 nm crosslinker in NaK2K, it is a striking demonstration of the potential for manipulation of selectivity by physically manipulating the SF conformation.

## TREK2 as a mammalian K[+] selective model

The remarkably similar ion-dependence of SF conformational dynamics unveiled by smFRET in KirBac1.1[21,22,31,44] and NaK2K (Figs. 1, 2)

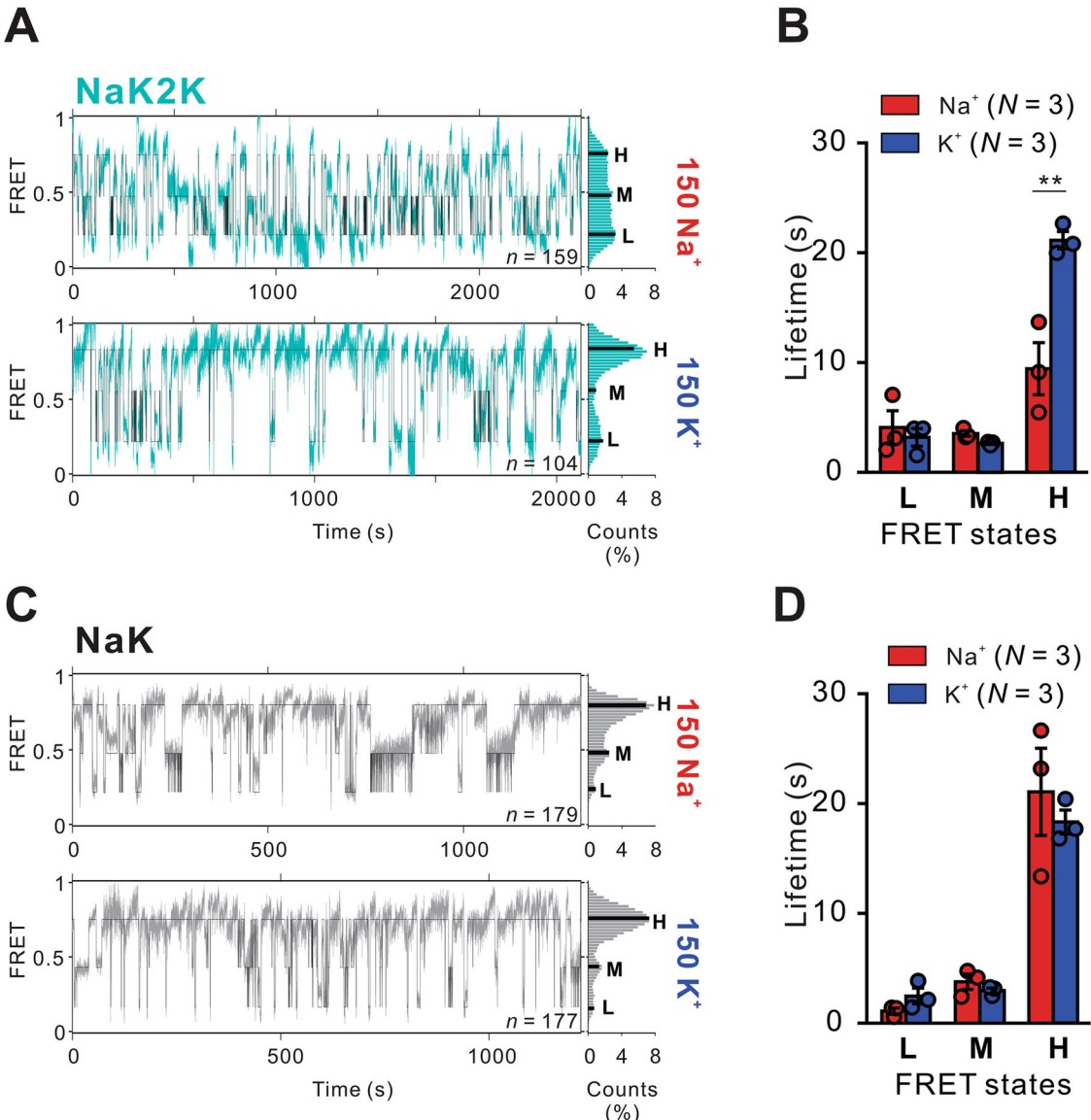

**Fig. 2 | Permeant ion-dependent kinetics of NaK2K and NaK SF. A** Concatenated (blue) and idealized (black) smFRET trajectories in K+ and Na+ for NaK2K. Histograms of experimental and idealized FRET distributions are shown to the right. **B** Calculated Lifetimes from the idealized traces shows a decrease in high FRET states for NaK2K in Na+. K+ compared with Na+ FRET lifetimes (Low FRET $p = 0.6276$; Medium FRET $p = 0.0640$; High FRET $p = 0.0096$). **C** Concatenated (gray) and idealized (black) smFRET trajectories in K+ and Na+ for NaK. Histograms of experimental and idealized FRET distributions are shown to the right. **D** Calculated Lifetimes from the idealized traces do not show any ion dependence of FRET states for NaK. K+ compared with Na+ FRET lifetimes (Low FRET $p = 0.1521$; Medium FRET $p = 0.3376$; High FRET $p = 0.5376$). In (**B**, **D**), $N$ = number of sub datasets. Error bars indicate s.e.m. **$p < 0.01$ vs. Na+. Comparisons between Na+ and K+ analyzed by unpaired t-test. Source data are provided as a Source Data file.

are consistent with a common mechanism throughout the K+ channel superfamily. The demanding experimental requirements for smFRET as implemented, in particular the need to specifically label only two single cysteine residues within a tetrameric channel complex have thus far precluded viable experiments on eukaryotic K+ channels. So-called K2P channels are a relevant eukaryotic K+ channel family of clinical importance, contributing to background K+ conductances in nearly all human cells, and representing attractive therapeutic targets for treatment of pain, migraine and various heart and lung disorders[45–49]. A potential major advantage of K2P channels for smFRET measurements is the ease of introducing two fluorophores per channel, which is greatly facilitated by the intrinsic subunit dimerization (Fig. 5B), although purification and labeling with maleimide fluorophores of eukaryotic membrane proteins still presents significant challenges, including for K2P channels, with several endogenous cysteines that

must be removed. We substituted three cysteines in the TREK2 crystal construct with alanines (Fig. 5A), and then purified, as previously described, from *P. Pastoris*[9,10,25,50–52] (See Methods). This cysteine-less TREK2 was functional, with essentially identical properties to WT either in PE:PG (3:1) or in asolectin lipids (Supplementary Fig. 4A, B), and was also successfully recombinantly expressed and purified with similar yield.

**The TREK2 SF reveals characteristic ion-dependent dynamics**
As with NaK and NaK2K, we applied smFRET to TREK2 channels by engineering, expressing, labeling, and reconstituting cysteine-less TREK2 constructs that contained only 2 cysteines (at position S105 in the extracellular region of the channel near the SF) within the dimeric channel (Fig. 5B, see Methods). Dimer fractions of Alexa Fluor 555 and Alexa Fluor 647 c2 maleimide-labeled TREK2 were collected by

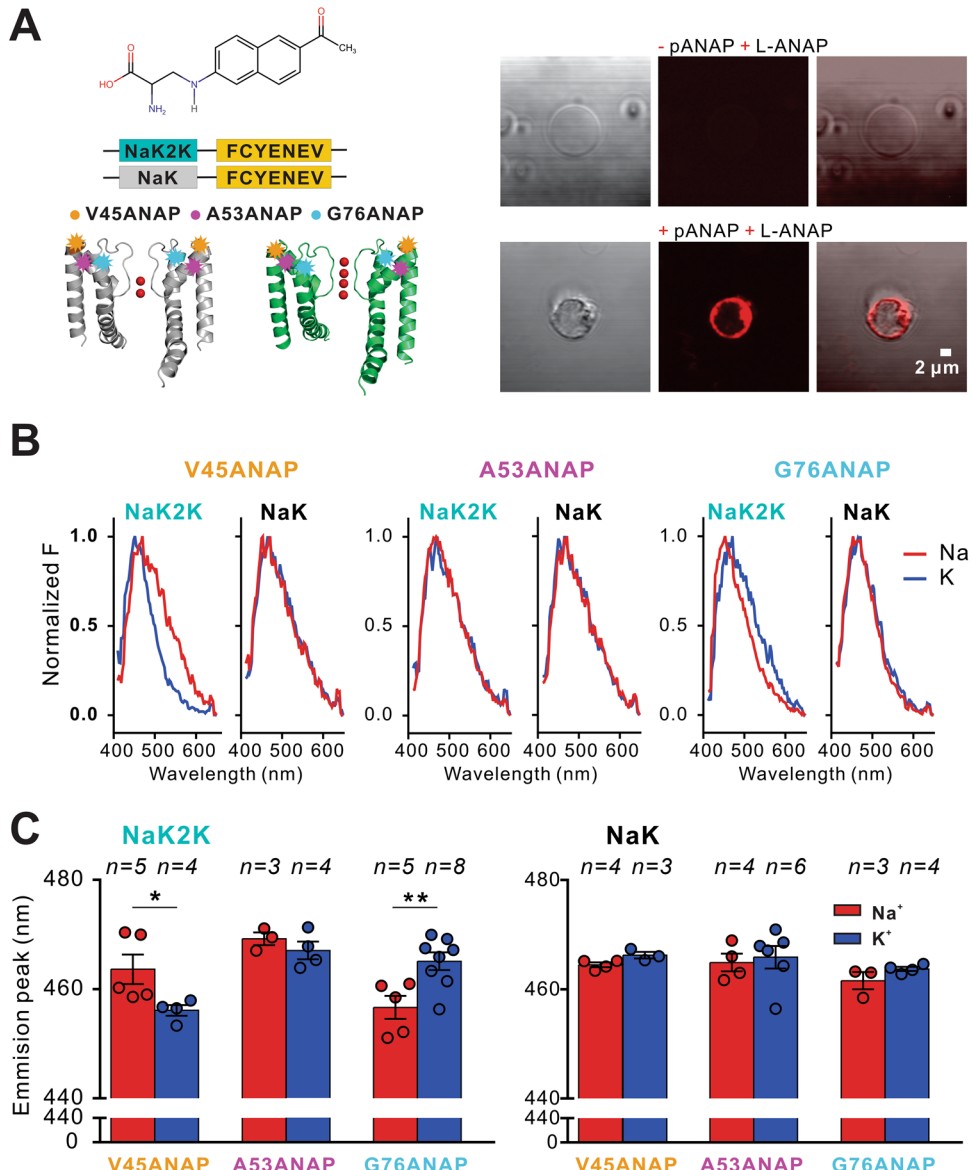

**Fig. 3 | NaK2K SF conformational changes in cellular membranes. A** Chemical structure of ANAP inserted using the amber codon at positions V45 (TM1), A53 (PH) and G76 (TM2) in NaK and NaK2K (PBD 3E8H and 3OUF, respectively, with C-terminal FCYENE sequence motif expressed in CosM6 cells (*left*) and representative (for at least *n* = 3 vesicles in each case) images of negative control giant plasma membrane vesicles (GPMVs) where the pANAP vector was not added, and positive GPMVs expressing ANAP-incorporated NaK (*right*). Pseudocolors for ANAP were used. **B** Representative emission spectra (red, 150 mM NaCl bath solution; blue, solution containing 150 mM KCl) of ANAP-incorporated NaK and NaK2K mutants. **C** Summary of shifts in ANAP emission peak at different incorporation sites (for each ANAP-incorporation site, *n* patches are indicated in each condition) in 150 mM Na⁺ or K⁺. Error bars indicate s.e.m. *p < 0.05 vs. Na⁺ and **p < 0.01 vs. Na⁺. Comparisons between Na and K analyzed by unpaired t-test. K⁺ compared with Na⁺ ANAP emission peaks (NaK2K V45ANAP *p* = 0.0492; NaK2K A53ANAP *p* = 0.3589; NaK2K G76ANAP *p* = 0.0081; NaK V45ANAP *p* = 0.0558; NaK A53ANAP *p* = 0.7479; NaK G76ANAP *p* = 0.1863). Source data are provided as a Source Data file.

size exclusion chromatography and reconstituted into liposomes (POPE:POPG = 3:1). Functional assays confirm that all labeled mutants retain channel activity (Supplementary Fig. 4C).

Single-molecule imaging was performed with a TIRF microscope as for NaK mutants. The FRET efficiency amplitude distributions of the TREK2 SF (Fig. 5C) show clear peaks, in both high K⁺ and high Na⁺, again requiring a minimum sum of three Gaussians for adequate fitting (with FRET efficiency peaks at ~0.15, ~0.35 and ~0.7). As found in NaK2K and KirBac1.1[21,23], the TREK2 SF shows predominantly stable high FRET configurations in 150 mM K⁺ (Fig. 5C), but more dynamic and less constrained, lower FRET efficiency, signals in 150 mM Na⁺. At this S105C location, there was also a strong donor-acceptor cross-correlation with a decay time constant of 0.86 s in Na⁺, but not in K⁺ (Supplementary Fig. 4D). Exponential fits to the lifetime distributions of

idealized concatenated FRET records (Fig. 5D, see Methods) again show a significant dependence on ionic conditions, with H state lifetime increasing, as Na⁺ is replaced by K⁺ (Fig. 5E). Finally, the FRET signal at the S105C position was sensitive to the presence of ML 335, a TREK SF stabilizer and activator[25], which showed a shift to higher FRET states in 100 µM ML 335 in high Na⁺ (Supplementary Fig. 4E).

## Discussion

### K⁺-induced formation of the K⁺-selective SF conformation is common to pro- and eukaryotic K⁺ channels

The signature TxGYG amino acid sequence in the K⁺ channel SF-loop is highly conserved throughout the K-channel superfamily, and a canonical and apparently stable structure of the SF, with 4 K⁺ ion binding sites formed by the backbone carbonyls of the GYG residues is seen in

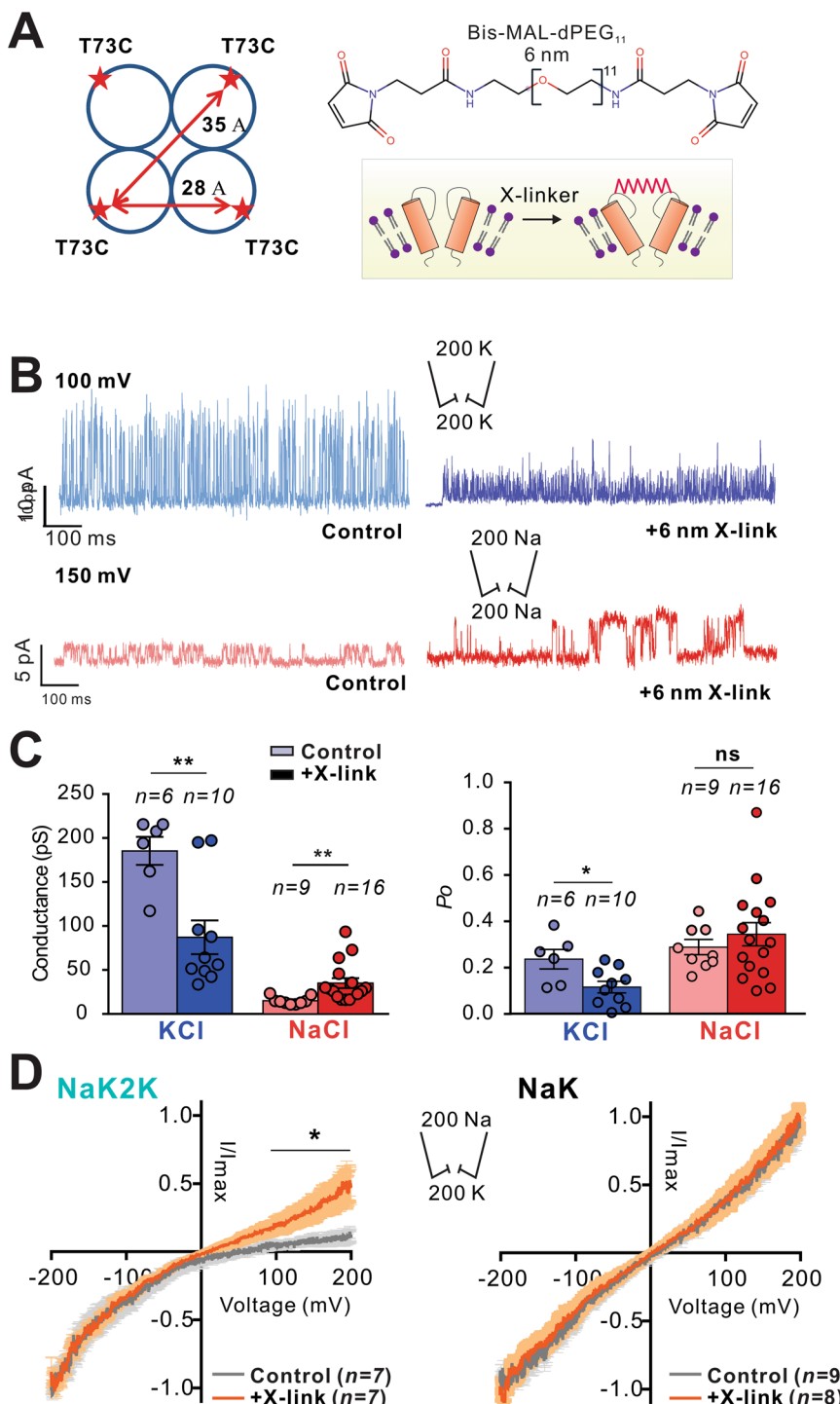

**Fig. 4 | Restricting extracellular SF flexibility reduces ion selectivity of NaK2K.**
**A** Cartoon representation of the proposed crosslink location between T73C residues (top of TM2) in NaK2K using a 6 nm crosslinker to restrict the extracellular SF flexibility. **B** NaK2K T73C single-channel recordings in symmetrical 200 mM K$^+$ (blue) and symmetrical 200 mM Na$^+$ (red) without and with the crosslinker (light and dark color, respectively). **C** NaK2K single-channel K$^+$ conductance is reduced after crosslinking (blue bars), whereas Na$^+$ conductance is increased (red bars). NaK2K open probability in K is reduced after crosslinking (blue bars). Crosslinked samples compared with Control (K$^+$ conductance $p = 0.0033$; Na$^+$ conductance $p = 0.0035$; Po in K$^+$ $p = 0.0194$; Po in Na$^+$ $p = 0.3688$). **D** Current-voltage relationships for multi-channel recordings in asymmetrical 200 mM K$^+$ (bath) and 200 mM Na$^+$ (pipette) without and with the 6 nm crosslinker shows significant left shift of reversal potential with the crosslinker, reflecting reduced ion selectivity in NaK2K. Error bars indicate s.e.m. in (**C**, **D**). In (**C**), \*\*$p < 0.01$ vs. Control. \*$p < 0.05$ vs. Control. ns (not significant), $n$ patches are indicated in each condition. Comparisons between control and crosslink conditions analyzed by unpaired t-test. Source data are provided as a Source Data file.

essentially all crystallized K$^+$ channel structures[2-14]. Very stable, high efficiency, smFRET signals obtained from the outer mouth of the SF in KirBac1.1 in high [K$^+$] conditions[21,22] are consistent with this seemingly innate arrangement. However, our previous studies showed that in the presence of very low [K$^+$], or when the SF is mutated to a non-selective channel, the smFRET-reported SF conformations in KirBac1.1 are transformed from stable and constrained to dilated and highly dynamic states[21,22]. These experiments provide compelling evidence

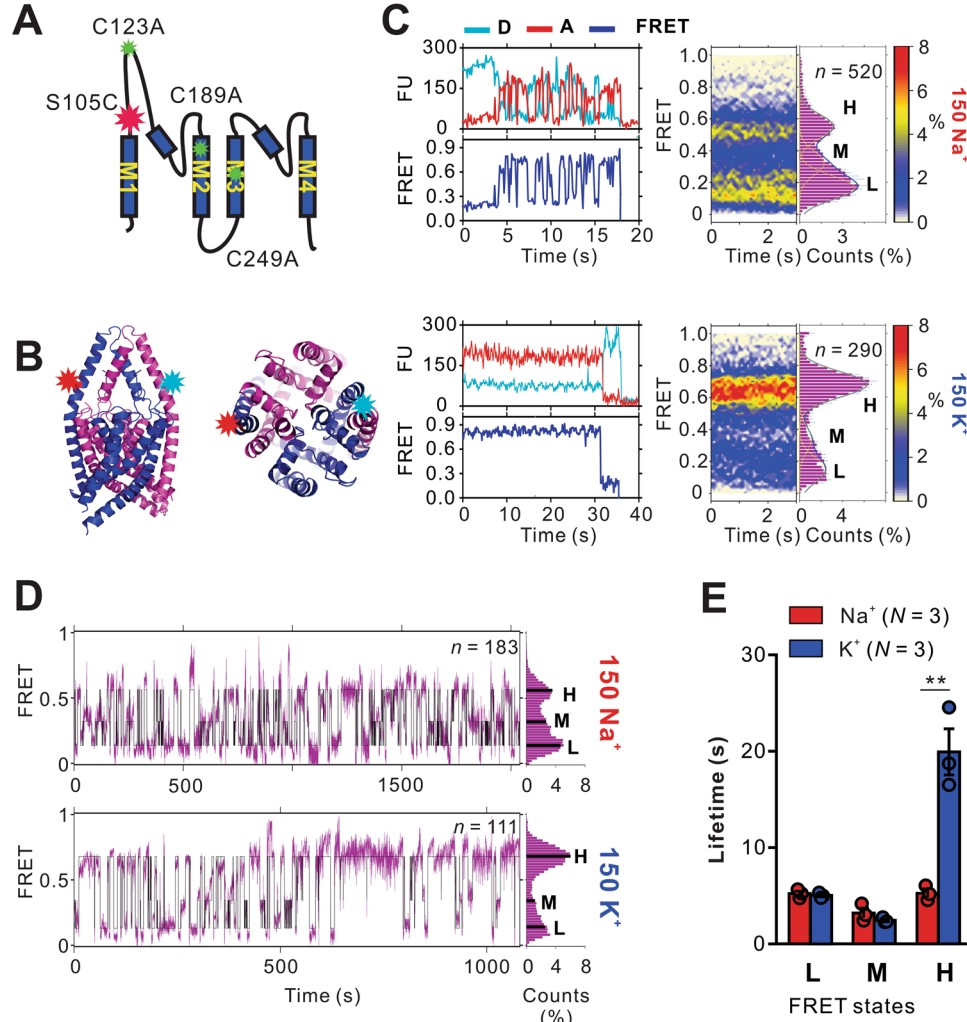

**Fig. 5 | Ion-dependent selectivity filter in TREK2. A** Cysteine to Alanine muta-tions to obtain cysteine-less TREK2 mutant. **B** Strategy for introducing two free cysteines within the dimeric TREK2 cysteine-less channel. The cysteine is intro-duced at position S105 at the top part of the first subunit (S1) TM1 domain. Domain organizations of the resulting proteins labeled with Alexa Fluor 555/647 c2 mal-eimide pair (PBD 4BW5). **C** Representative smFRET traces for TREK2 S105C, as well as FRET contour plots and histograms ($n = 520$ and 290 TREK2 traces, in the pre-sence of 150 mM NaCl or KCl, respectively). **D** Concatenated (mauve) and idealized (black) smFRET trajectories in K$^+$ and Na$^+$ for TREK2. Histograms of experimental and idealized FRET distributions are shown to the right. **E** Calculated Lifetimes from the idealized traces shows a decrease in high FRET states for TREK2 in Na$^+$. K$^+$ compared with Na$^+$ FRET lifetimes (Low FRET $p = 0.5951$; Medium FRET $p = 0.2164$; High FRET $p = 0.0038$). In blue, D, donor and in red A, acceptor. $n$ = number of molecules. $N$ = number of sub datasets. Error bars indicate s.e.m. **$p < 0.01$ vs. Na$^+$. Comparisons between Na$^+$ and K$^+$ analyzed by unpaired t-test. Source data are provided as a Source Data file.

that the SF-loop conformation is actually dependent on K$^+$ ion occu-pancy within the SF and that, rather than being a pre-existing state that permits K$^+$-selective permeation, the constrained SF-loop conforma-tion is actually generated and maintained by the K$^+$ ions themselves. The present study provides key evidence that this is indeed a general phenomenon, relevant to the whole K$^+$ channel superfamily. The data not only show that there is essentially identical ion-dependence of smFRET signals in both prokaryotic NaK2K and eukaryotic TREK-2 K$^+$-selective channels, but also that reversion to the non-selective TVGDGN sequence of NaK channel leads to a SF-loop conformation that is completely insensitive to the ionic conditions, as in KirBac1.1[21,22].

High-resolution cryoEM and computer simulations have revealed additional dilated SF structures of the activated and C-type inactivated Shaker Kv channel and its W434 mutant in lipid bilayers[53]. A recent study reported structures of Kv1.3 channels that showed a dynamic selectivity filter with two discernable dilated conformations[54], the authors hypothesizing that these may correspond to partially inacti-vated and inactivated states of the channel. Very consistent with our FRET data, a recent study using solid-state NMR (ssNMR) showed that

the SF of K$^+$-selective NaK2K becomes dynamic when K$^+$ is replaced by Na$^+$[55], while non-selective NaK has multiple stable selectivity filter conformations that adapt to the presence of either K$^+$ or Na$^+$ ions[56]. It is intriguing that, despite losing ion dependence of the FRET signals in the non-selective NaK, the FRET efficiency is stably high in both Na$^+$ and K$^+$, rather than dynamic, predominantly low FRET efficiencies, as are seen in non-selective KirBac1.1 mutants[21]. We speculate that this is a consequence of the underlying structure of NaK, where the latest high resolution microED and crystal structures[57,58] have shown that there is an external ion binding site for either Na$^+$ or K$^+$ at position Asn68 (Asp in NaK2K), which might stabilize the external part of the SF loop in NaK, whether in Na$^+$ or K$^+$ solutions.

### Physiologic and pharmacologic relevance of ion-dependent SF conformation

Since the K$^+$-stabilized high FRET state is not saturated at even very low [K$^+$], SF dynamics should be present under physiological, ionic conditions. This possibility, suggested by Zhou et al[3]. has not been detected, and may be difficult to capture, in crystal or Cryo-EM

structures. Assuming isotropy, and an orientation factor $\kappa^2 = 2/3$, the predicted distances corresponding to FRET efficiencies ~0.8, ~0.45 and ~0.2 in NaK and NaK2K are 40, 52 and 64 Å, and for FRET efficiencies ~0.7, ~0.35 and ~0.15 in TREK2 are 44, 56 and 68 Å, respectively. NaK, NaK2K and TREK1-2 have been crystallized multiple times, and the predicted T73 Cα-Cα distances for both NaK[18] amd NaK2K[27,43] in high [K⁺] or high [Na⁺], as well as the predicted S105 (or TREK1 equivalent S75) Cα-Cα distances in high [K⁺][10,25] or high [Na⁺][25] are all 35–38 Å, very close to those predicted by the highest FRET efficiencies in our experiments. Lack of any crystal structures with more dilated SF conformations might then be taken as evidence against any physiological relevance of the dilated states. However, since proteins will crystallize in minimum energy conformations, and since removal of permeant K⁺ ions destabilizes K⁺ channel proteins outside of cell membranes[59–61], conceivably SF expansion obviates stable crystals or stable cryo-EM conformations. The demonstration of ion-dependence of SF conformation of NaK2K, but not of NaK, in living cells, using L-Anap imaging, is thus an important confirmation of physiological relevance of SF conformational dynamics, consistent with our previous demonstration that the Kir-Bac1.1 SF conformation does transit between stable constrained, and dynamic, dilated states in physiologically relevant ion gradients[21].

The potential that pharmacological agents interfering with the SF might alter selectivity or gating of the channel, rather than simply blocking permeation is a relatively novel concept. Proks et al.[30] have recently shown that NFx inhibition of TREK2 can be influenced by agonists which alter SF stability, such as ML335, as well as by an intrinsic voltage-dependent gating process within the SF. Lolicato et al.[25] have shown that ML335 stabilizes the TREK1 SF, showing ions present in S1-S4 regardless of the K⁺ concentration, potentially explaining the shift to higher FRET states that we observe when TREK2 is incubated with ML 335 in high Na⁺. Furthermore, Decher et al.[48] have presented a pharmacological strategy to rescue a selectivity defect of the TREK-1 pore using BL-1249. Our finding that covalent attachment of a 6 nm chemical crosslinker across the outside of the SF, predicted to force SF widening, results in reduced K⁺ conductance and enhanced Na⁺ conductance in NaK2K provides further direct evidence for effects of external conformation controlling K⁺-selectivity, and demonstrates a novel potential way to manipulate ion selectivity by expanding or constraining SF motions.

# Methods

## Plasmids
Plasmids containing monomeric and tandem dimeric NaK and NaK2K Δ19 protein-encoding cDNAs were constructed using the pET28a expression vector (Kanʳ)[21]. For tandem dimeric constructs, a short GGGSGGGS linker was introduced between the two copies of NaK or NaK2K coding DNA and a His₈ tag was introduced in the C termini of all protomers, for metal affinity purification. Plasmids containing TREK2 WT and cysteine-less (C123A, C189A, C249A mutated) from Gly67 to Glu340 protein-encoding cDNAs, with a C-terminal purification tag with a tobacco etch virus (TEV) protease cleavage site, a 10x His purification sequence and a FLAG tag were constructed in the pPICZ expression vector[8,10]. For electrophysiological recordings in COS cells, NaK and NaK2K Δ19 were subcloned into a modified pCEU vector containing a C-terminal GFP-His₈-FCYENE tag plasmid and for L-Anap experiments the same constructs without GFP were used. All mutations were introduced by QuikChange II XL site-directed mutagenesis kit (Agilent Inc.) and confirmed by DNA sequencing. Oligonucleotides used for mutagenesis are listed in Source Data file.

## Protein expression, purification, and fluorophore labeling
Monomeric and tandem dimeric NaK proteins were expressed in E. coli and and purified following standard protocols[21,27]. The metal affinity-purified proteins were passed through a Superdex-200 10/300 size exclusion column (GE Healthcare Inc.) with running buffer containing 20 mM Hepes, 150 mM KCl, 5 mM DM, pH 7.5 for NaK. Tetrameric fractions were pooled and concentrated via Amicon Ultra-4 centrifugal filter (MWCO 50 KDa, Millipore Inc.). TREK2 proteins were expressed in *Pichia Pastoris* and purified following standard protocols[8,10,50]. The metal affinity-purified proteins were passed through a Superdex-200 10/300 size exclusion column (GE Healthcare Inc.) with running buffer containing 20 mM Tris, 150 mM KCl, 2 mM DDM, pH 8 for TREK2. Tetrameric fractions were pooled and concentrated via Amicon Ultra-4 centrifugal filter (MWCO 50 KDa, Millipore Inc.). Fluorophore labeling was started immediately after gel filtration by adding 1:1 (molar ratio) mix of Alexa Fluor 555 and 647 c2 maleimide to protein solution at final protein:fluorophore molar ratio of 1:5. Labeling reactions proceeded at room temperature for 1 h and were then terminated by addition of 2-mercaptoethanol at final concentration of 10 mM. A second metal affinity purification was performed to remove free fluorophores or those associated with protein through non-covalent bonds. The labeled proteins were loaded onto a size exclusion column (Superdex-200 10/300, GE Healthcare Inc) and tetrameric fractions were collected and concentrated for liposome reconstitution. A labeling control using protein without intrinsic cysteine was always included to evaluate fluorophores bound non-specifically or associated with protein through non-covalent bonds. All the purifications were performed under 4 °C except for labeling reactions.

## Protein reconstitution
POPE (1-palmitoyl-2-oleoyl-sn-glycero-3-phosphoethanolamine) and POPG (1-palmitoyl-2-oleoyl-*sn*-glycero-3-phospho-(1′-*rac*-glycerol)) lipids (3:1, w/w) were dissolved in buffer containing 20 mM Hepes, 150 mM KCl and 30 mM CHAPS, pH 7.5 at final concentration of 10 mg/ml. Protein labeled with Alexa Fluor 555 and 647 fluorophores was mixed with lipid solution at protein:lipid ratio of 1:200 (w/w), with 2% biotinylated-POPE (w/w, of the total lipids). The lipid/protein mix was incubated at room temperature for 20 min then passed through a sephadex G-50 desalting column to remove detergents, thereby forming proteoliposomes. Residual detergents were removed by dialysis against 1 L buffer containing 20 mM Hepes, 150 mM KCl or NaCl, pH7.5 for 3 times and proteoliposomes were harvested and stored at −80 °C freezer for single-molecule imaging[21,22,44].

## Single-molecule imaging
Chamber slides were prepared following the protocol of Joo et al.[62]. An objective-based TIRF microscope, built on a Nikon inverted microscope (TE-2000s) with 100× APO TIRF NA1.49 objective lens, 532 nm and 640 nm lasers, was used for single-molecule imaging. Donor and acceptor emissions were separated by OptoSplit II (Cairn Inc.) with 638 nm long pass beam splitter, passed through 585/65 nm and 700/75 nm emission filters (Chroma Inc.) and then collected by Evolve 512 delta EMCCD camera (Photometrics Inc.). A CRISP autofocus system (ASI Inc.) was incorporated to compensate for focus drift due to mechanical vibrations and thermal fluctuations. Liposomes containing fluorophore-labeled proteins were immobilized on the slide surface by biotin-neutravidin interactions with biotinylated-POPE in liposomes. Fluorophores were excited by 532 nm laser, and movies were collected using NIS-element (Nikon Inc.) with frame rates of 10 per second (i.e., time resolution of 100 ms). Laser power was ~9.1 W/cm² (at the objective lens side). Recording times were 2 min. Except for the different concentrations of cations, all imaging buffers contained ~3 mM 6-hydroxy-2,5,7,8-tetramethylchroman-2-carboxylic acid (Trolox), 2 mM 4-nitrobenzyl alcohol (NBA), 2 mM cyclooctatetraene (COT), 5 mM protocatechuic acid (PCA) and 15 μg/μL protocatechuate-3,4-dioxygenase (PCD) to enhance the photostability of the fluorophores[63,64]. 50 μM β-escine was use to permeabilize liposomes[21,65] for experiments with symmetric ionic conditions. Control liposomes reconstituted with labeled control protein at the same

concentration were included to evaluate the fluorescent impurities, ensuring that they were less than 5% in comparison with sample liposomes. For every protein, at least two independent labeled samples were used; for every sample, ≥10 videos were collected.

## Single-molecule imaging data analysis

For every video, individual molecules were identified, and donor and acceptor fluorescence intensity profiles were extracted by IDL scripts developed by the Ha group[66,67]. Leak and direct excitation corrections were not applied, as leakage was <0.06, and direct excitation was undetectable. Traces were inspected and selected manually following established criteria[21,22,31,44]. The bin size of all time histograms was set as 0.025 of recording time, ensuring an equal contribution from each trace to avoid dominant effects of long traces[32]. FRET contour plots were generated from the first 3 s of each trace. For idealizing smFRET traces using HaMMy software[68], FRET traces in the same condition used to make the histograms were concatenated into a single file. For the ML 335 experiments, the smFRET traces were extracted and preselected using the Autotrace function of the SPARTAN software, with criteria set as FRET Lifetime >50 frames, donor/acceptor correlation coefficient between −1.1 and 0.5, signal-to-noise ratio >8, Cy3 blinks <4, and overlap molecules removed[69]. The resulting traces were further picked manually, following established criteria[21,22,31,44]. Subsequent analysis was performed using Matlab, Microsoft Excel, and GraphPad Prism 6.

## Fluorescence liposome flux assay

Proteins were reconstituted into POPE/POPG liposomes at protein:lipid ratio of 1:500 for NaK and different protein:lipid ratio for TREK2, by passing through Sephadex G-50 desalting columns equilibrated with buffer 20 mM Hepes, 150 mM KCl, pH7.5. Immediately before flux assay, the extraliposomal buffer was replaced by buffer containing 150 mM NMDG pH7.5. ACMA stock was then added to reach a final concentration of 13 μM into a 96-well plate. Baseline fluorescence (excitation wavelength 400/30 nm and emission wavelength 495/10 nm, Top50 Mirror) was measured by a Synergy 2 plate reader[21,22]. All flux data were normalized to the maximum quenching after valinomycin addition.

## L-Anap incorporation

L-ANAP was purchased from Cayman chemicals. pANAP vector was purchased from Addgene. ANAP was incorporated into NaK or NaK2K-FCYENE protein by introducing a TAG amber stop codon mutations at positions 45, 53 and 76. During transfection, 0.7 μg pANAP vector was co-transfected with 2 μg plasmid of NaK mutants. 20 μM ANAP was directly mixed into the culture medium. COS-M6 cells media was change after 48 hours to remove the L-Anap excess during 24 h. 72 h after transfection the cells expressing ANAP-incorporated NaK channels were ready to generate GPMVs.

## GPMV generation

GPMVs were prepared using established protocols[41,42]. In brief, 2−3 days after cell transfection, the cells were washed three times with phosphate buffered saline (PBS), followed by being washed twice with GPMV buffer (2 mM CaCl₂, 10 mM Hepes, 150 mM NaCl, pH 7.4). Later, the cells were incubated with GPMV vesiculation buffer (25 mM paraformaldehyde (PFA) and 2 mM dithiothreitol (DTT) in GPMV buffer) at 37 °C, 200 rpm, during 2−3 h. After the incubation, GPMVs that had detached from the cells were gently decanted into a conical tube. The conical tube was placed in a 4 °C refrigerator to allow the GPMVs to settle down at the bottom of the tube for further use without any purification treatment.

## L-Anap hyperspectral imaging

L-Anap fluorescence images of GPMVs using a bath solution containing 150 mM NaCl or KCl, 20 mM Hepes and pH 7.5, were acquired using a

Zeiss LSM-880 two-photon microscope with a Plan-Apochromat 63× 1.4 NA oil immersion objective at 760 nm excitation. The entire emission spectrum of L-Anap was recorded using the LSM-880 spectral detector mode from 410 to 650 nm at a spectral resolution of 3 nm. The recorded emission spectra were analyzed using FIJI and the maximum were measured by fitting to the skewed Gauss equation in Origin.

## DiBac fluorescence assays

72 h after transfection with NaK WT or NaK L-Anap-incorporated mutants, COS-M6 cells where loaded during 30 min using a solution containing (in mM): NaCl or NMDG 136, KCl 1, CaCl₂ 2, MgCl₂ 1, HEPES 10, glucose 10, and DiBac 0.03 with pH adjusted to 7.4 with NaOH. DiBac fluorescence images were acquired using the Zeiss LSM-880 two-photon microscope with a Plan-Apochromat 20× 0.8 NA objective, at an excitation wavelength of 488 nm, detected at an emission range of 500 − 580 nm. The recorded DiBac cell fluorescences were analyzed using FIJI. Every data point is an individual image with the mean fluorescence of at least 20 cells.

Western blot assays. After purification, NaK and NaK2K T73C (with F92A) samples were incubated for 1 h in 10 mM crosslinker (3 nm or 6 nm). Due to the intrinsically high NaK2K tetramer stability, NaK2K samples were destabilized by further incubation in 5.3 M guanidinium hydrochloride for 16 h at room temperature. For immunoblotting, an appropriate volume of 4 × Laemmli sample loading buffer was added to the sample and loaded onto 4−20% gel (Bio-Rad). Protein subunits were separated by electrophoresis in 25 mM Tris Base, 190 mM Glycine, 0.1% SDS running buffer for 1 h at 150 V for NaK and 2.5 h at 50 V for NaK2K. Proteins were transferred to PVDF membrane (Sigma) and membrane blocked in 5% (w/v) milk in TBST buffer (0.2 M Tris, 1.37 M NaCl, 0.2% Tween-20, pH 7.4) at room temperature for 1 h. Blots were incubated with primary antibody conjugated with HRP against the C-terminal Histidine tag at room temperature for 2 h (6x-His Tag Antibody, MA1-21315-HRP, ThermoFisher,1:5,000 dilution). Membranes were washed three times and imaged by chemiluminescence (Pierce) by using a Chemidoc imaging system (Bio-Rad). The images were further analyzed for band intensities using GelAnalyzer software.

## Electrophysiological recordings

All NaK mutant proteins for electrophysiology assays contained an extra Phe92 to Ala mutation to enhance the single-channel conductance[27]. CosM6 cells were transfected with 2 μg of NaK constructs using FuGENE6 (Promega) and used for patching within 2−3 d after transfection. Whole-cell currents were recorded using an Axopatch 200B amplifier and Digidata 1200 (Molecular Devices). Recordings were sampled at 5 kHz and filtered at 1 KHz. Currents were measured from −140 to 140 mV with a 10 mV increments and a holding potential of −60 mV in high-Na⁺ bath solution containing (in mM): NaCl 134, KCl 6, CaCl₂ 2, MgCl₂ 1, HEPES 10, and glucose 10, with pH adjusted to 7.4 with NaOH. The pipette solution contained (in mM) potassium gluconate 130, KCl 10, MgCl₂ 1, HEPES 10, CaCl₂ 0.5, K₂HPO₄ 4, and EGTA 5, with pH adjusted to 7.2 with KOH.

Single- and multi-channel recordings were obtained by patch-clamping reconstituted proteoliposomes. The protocol is based on methods used with reconstituted MthK in soybean polar lipids[70]. A total of 25 mg/ml lipids (Avanti) in chloroform were dried under argon and kept overnight in vacuum. The dried lipids were resuspended using bath sonication in 250 mM KCl, 30 mM Hepes, pH 7.5, and 0.1 mM CaCl₂ to a final concentration of 15 mg/ml. To the suspension, 5 mM DM (NaK) or 0.5 mM DDM (TREK2) were added so that the protein:lipid molar ratio was either 1:1000 or 1:2500 for single- and multi-channel recording studies. Detergents were removed by O/N dialysis using 25 KD cutoff Slide-A-Lyzer Dialysis cassette. The dialysis buffer was refreshed next day, and after 4 hr, the proteoliposomes were aliquoted and stored at −80 °C. 30 μl of proteoliposomes were

placed on a clean glass slide and dried in a desiccator under vacuum at 4 °C. The sample was then rehydrated with 50 µl buffer (250 mM KCl, 30 mM Hepes, pH 7.5, and 0.1 mM CaCl$_2$) for >2 hr, which yielded giant multilamellar vesicles (GMV). For NaK single-channel recordings the pipette solution contained 200 mM KCl or NaCl, 10 mM Hepes, 1 mM EGTA, pH 7.4 buffered with KOH or NaOH. The standard bath solution contained 200 mM KCl or NaCl, 10 mM Hepes, 1 mM EGTA, pH 7.4 buffered with KOH or NaOH. For multi-channel recordings the pipette solution contained 200 mM NaCl, 10 mM Hepes, 1 mM EGTA, pH 7.4 buffered with NaOH. The standard bath solution contained 200 mM KCl, 10 mM Hepes, 1 mM EGTA, pH 7.4 buffered with KOH. For cross-linking experiments 1 mM of the 3 or 6 nm crosslinkers (Bis-Mal-dPEG3, Bis-Mal-dPEG11, Quanta Biodesign) was added to the pipette and the bath solutions. Liposomes were incubated in the bath chamber for at least 5 min in the presence of the crosslinker. After patch excision and before recording, the sample was incubated another 5 min to allow the crosslinking reaction. For TREK2 recordings the pipette solution contained 200 mM KCl, 10 mM Hepes, 0.5 mM CaCl$_2$, pH 7.4 buffered with KOH. The standard bath solution contained 200 mM KCl, 10 mM Hepes, 0.5 mM CaCl$_2$ and 200 µM TPA, pH 6 buffered with KOH. Recordings were made and digitized with the Axopatch 1D patch-clamp amplifier and the Digidata 1200 digitizer (Molecular Devices). Data were collected at 10 kHz and analyzed with the pClamp software suite (Molecular Devices). The pipettes with at least 5 M Ohm were fabricated from the Kimble Chase soda lime glass with a Sutter P-86 puller (Sutter Instruments). All measurements were carried out at 150 mV (except for NaK2K and TREK2 in symmetrical K$^+$ that were carried out at 100 mV) membrane potential.

## Statistics

The FRET histogram data for each sample/condition were presented as equal contributions from all individual molecule traces (rather than each data point). All selected traces were >3 s, and the FRET contour map for each sample/condition was calculated from the first 3 s of all traces, with equal contributions from all individual traces. The trace number $n$ for each sample/condition is included in the accompanying figure legends. For each condition, FRET data were split into three different datasets with a similar number of traces coming from multiple different videos in each case, to assess variability. The calculated rate constants and state probabilities are therefore represented as mean ± SEM of $n = 3$ datasets. To assess state dwell time distributions, datasets were used to fit monoexponential decay functions. Cross-correlation analysis was performed with smFRET sub datasets maintaining the original full set histogram distributions. Energy landscapes were plotted using a mixture of smFRET peak occupancies and rate constants calculated. All data are presented as mean ± SEM.

## Reporting summary

Further information on research design is available in the Nature Portfolio Reporting Summary linked to this article.

## Data availability

All data generated or analyzed during this study are included in this article (and the Supplementary Information files), and are available from the corresponding authors upon reasonable request. Previously published PDB structures references in this work can be found under accession codes 4BW5 (TREK2), 3E8H (NaK), and 3OUF (NaK2K). Source data are provided with this paper.

## Code availability

The script that converts.tif movie files to.pma files for later data processing is available upon request. IDL scripts developed by the Ha group (fully described in refs. 66,67) were used for movie data processing (.pma files). Subsequent analysis was carried out using Microsoft Excel, Matlab and GraphPad Prism 6.

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

## Acknowledgements

We are very grateful to Dr. Stephen J. Tucker (Oxford University, UK), for providing the cysteine-less TREK-2 construct. We thank Drs. Sun-Joo Lee, Shizhen Wang, Grigory Maksaev, and PhD. student Jian Gao for many helpful discussions and experimental assistance. We also thank Drs. Baron Chanda, Willy Carrasquel-Ursulaez and Vinay Idikuda for experimental assistance in giant liposome reconstitution. The work was funded by NIH grant R35 HL140024 (to CGN) and by a Postdoctoral Fellowship from The McDonnell Center for Cellular and Molecular Neurobiology, Washington University in Saint Louis (to MM).

## Author contributions

M.M. and C.G.N. conceived and designed the studies; M.M. and X.W.N. performed the L-Anap imaging, assisted by D.W.P.; J.B.B. subcloned TREK2 in pPICZ vector and helped with the TREK2 expression; MM performed the rest of the research; M.M. analyzed the data with help from C.G.N. and X.W.N.; M.M. and C.G.N. prepared the paper with inputs from all authors.

## Competing interests

The authors declare no competing interests.
