## [Peer Review File · Nature Communications]

Conformational plasticity of NaK2K and TREK2 potassium channel selectivity filters

Editorial Note: Parts of this Peer Review File have been redacted as indicatedReviewers' Comments:

Reviewer #1:

Remarks to the Author:

In this manuscript Matamoros et al. describe selectivity filter dynamics in the nonselective ion channel NaK, the K⁺ selective NaK mutant NaK2K, and the two pore eukaryotic channel TREK2. Following their prior work on KirBac1.1, the authors introduce cysteine mutants in the selectivity filter loop of both NaK and NaK2K. Fluorophores are attached to these sites and the single molecule FRET (smFRET) signal is measured in buffers containing 150 mM K⁺ or Na⁺. These smFRET measurements show that the selectivity filter of NaK2K is far more dynamic and less stable in a Na⁺ buffer compared to NaK. The smFRET data shows no ionic dependence. The authors then interrogate these channels in biological membranes by incorporation of L-Anap in COSm6 cells. These fluorescence measurements are consistent with the smFRET results. They then confirm these selectivity filter features are also observed in TREK2 using an extensive mutant (where native C residues are mutated to A) to allow for labeling specificity. This is a very nice study showing that selectivity filter dynamics and conformational states are important for most K⁺ channel classes, and not simply those undergoing C-type inactivation.

This is an extremely well written and well-organized manuscript. I find all the experiments convincing. The quality makes sense as this is the most recent in a series of FRET-based studies by the lab of Colin Nichols. My only criticism is that this study fits so well into this group's prior work that it does not feel entirely new. However, showing that sf dynamics extend beyond a prokaryotic Kir channel is important. I really like the new data showing that linkers restricting loop motions can abolish, to a degree, ionic selectivity. I think techniques of this type will be very important moving forward. I recommend publication in its current state.

Reviewer #2:

Remarks to the Author:

This report, entitled "Conformational plasticity of NaK2K and TREK2 potassium channel selectivity filters" is a thorough and well-designed study of the conformational dynamics of the potassium channel selectivity filter. The results presented in this work mirror a number of very recent structural studies of potassium channels, all of which converge on a theme, showing a dilation of the upper selectivity filter upon C-type inactivation or destabilization of the filter by removal of K⁺. The data in this study contribute to a growing convergent understanding of SF dynamics, though the observation that the non-selective NaK filter remains constricted is perhaps the most surprising result, as this data breaks with the overall trend showing dilation of the upper filter correlated with loss of K⁺ selectivity. The paper is clear and well written, the experimental techniques are appropriate for the questions being asked, and the results are intriguing, both for these specific channels and as an example of the use of this smFRET approach to explore SF dynamics in a range of ion channels. I have two major areas of identified weakness I hope the authors can address, and a few minor points that should be easily fixable.

1) The work by the authors showing that their findings in NaK2K are mirrored in the TREK2 channel are intriguing and consistent with crystallographic structures of TREK1 in high and low K⁺ (Manuscript Ref 25, Lolicato Sci Adv 2020). That being said, the S105 residue is a surprising choice for a probe of conformational changes within the selectivity filter. Whereas the T73 residue used for the smFRET studies of NaK2K is in a flexible loop region that borders the SF, TREK2 S105 is in the middle of the alpha helix that comprises TM1 and the helical cap above the membrane. S105 faces away from the selectivity filter in structural models of TREK2 and is approximately ~10-15 angstrom away from the nearest SF adjacent residue. While the choice of this position may have been necessitated by the need to find a cysteine accessible residue for modification, it becomes hard to directly attribute changes in smFRET at this distant site to changes in selectivity filter structure, rather than being simply a readout

of a more global conformational change in the protein.

a. To better understand the TREK2 results, I would ask the authors to perform an additional smFRET experiment with the TREK2 S105C construct in the presence of both 150 Na⁺ and the selectivity filter stabilizing drug ML335. If S105 does in fact report on the constricted versus dilated state of the filter, this drug should elucidate that. One would predict that ML335 should stabilize the SF in the conductive (and high FRET) state, counteracting the effect of 150mM Na⁺. This additional result would more closely link the S105 smFRET data to the structural state of the SF.

2) By using a PEG Maleimide crosslinker, the authors covalently trap a residue in the upper selectivity filter region in an extended state. Using single channel recordings, they demonstrate that the crosslinked NaK2K channels exhibit a decrease in single channel conductance for potassium and an increase in single channel conductance to sodium. I have several points to address in this section of the manuscript.

a. Data for these single channel recordings are somewhat incomplete. While changes in single channel conductance are reported, there is no discussion of other single channel parameters, most notably Po or open/closed channel dwell times. A more detailed analysis of these data would be helpful to understanding the results presented in this section.

b. The interpretation of the presented data states that the electrophysiological results are consistent with a change in ion selectivity for K⁺ versus Na⁺. However, the observed changes in conductance in experiments that utilize only mono-cationic solutions do not appear to conclusively support that assertion. The authors should more fully discuss their rationale for this claim. Alternatively, an additional experiment like the one presented in Supplemental Figure 2A, demonstrating a shift in reversal potential after reaction with the crosslinking reagent, would more definitely corroborate the claim of a shift in ion selectivity in the extended conformation after crosslinking.

c. From the methods, it appears that the PEG Maleimide was added to the bath and pipette solution during recordings, but the purified protein was not pre-incubated with the crosslinker. There is no description of the reaction rate of the PEG Maleimide with the channel or the time between application of the PEG Maleimide and initiation of the recordings. How do the authors differentiate between channels that have been effectively crosslinked from those that haven't? The wide distribution of the individual data points in Figure 4c suggest a heterogenous population of channels, perhaps consistent with incomplete modification of the T73C residue.

Minor Points –

1) Page 5, line 118 – The manuscript reads “If the trace data are very noisy, cross-correlation analysis between donor and acceptor fluorescence intensities can be performed.” The language here is a bit confusing. By noisy, do the authors mean instances where the FRET signal is dynamic?

2) Supplemental Figure 2c, A53An mutant. While not central to manuscript, I am confused by the results for this mutant in NMDG. With Na⁺ eliminated as the cation responsible for depolarization, this mutant still is significantly depolarized relative to untransfected cells. Is the assumption that this mutant also allows NMDG to permeate through the NaK2K channel?

3) Figure 3, panel A – Identifying the locations of the introduced mutations is somewhat difficult. I would suggest enlarging the structural models and revising the color coding in the models, as they do not appear to match the provided key.

4) The authors might also like to cite a recently published study on inactivated conformations of the Kv1.3 channel, with similarities to the presented results: Selvakumar, P., Fernández-Mariño, A.I., Khanra, N. et al. Structures of the T cell potassium channel Kv1.3 with immunoglobulin modulators. *Nat Commun* 13, 3854

Reviewer #3:

Remarks to the Author:

The manuscript by Matamoros et al. studies the conformation of the selectivity filter (SF) of two potassium channels, the bacterial NaK and the mammalian TREK2. Two versions of NaK are used, the WT and Na2K2 (obtained by mutation of the SF), which is selective for K⁺ over Na⁺. This is a high-

quality paper that uses interesting complementary approaches, including activity assays. However, I have reservations with regards to the impact/significance, and with some experimental components. The same laboratory previously (Refs 21 and 22 of the manuscript) studied by single-molecule (sm) FRET the SF of another type of K⁺ channel (KirBac1.1). These studies showed changes in the SF filter, which becomes more dynamic in the presence of Na⁺. The current work is somewhat confirmatory, as it shows similar observations for NaK2K and TREK2. This is still relevant because it suggests that this dynamic change could be a general behavior across a wide range of K⁺ channels. However, the current data it is not particularly ground-breaking, in particular because a different group performed ssNMR of NaK2K, providing data that showed the same general conclusions. These circumstances significantly reduce my enthusiasm for this otherwise strong and carefully done work.

The crosslinking experiments are the weak element of the manuscript. I have serious reservations that the use of the crosslinker is indeed providing the information that the manuscript claims. First of all, the crosslinking scheme allows for different types of crosslinking events; due to the presence of four Cys, a crosslinking pair could have a distance of 28 Å, or 35 Å. As a result, one can expect that the crosslinked samples contain an unknown mixture of crosslinked populations. More worrisome, more than one crosslinker could be engaged in a single channel. In these circumstances, a careful examination of the product of the crosslinking should be performed. However, the manuscript does not contain these studies. A simple SDS-PAGE should be able to 1) identify the presence of crosslinking, and 2) characterize the resulting samples (more advanced techniques would be welcome). Importantly, comparison between different experimental conditions (Na⁺ vs K⁺) would identify if comparable crosslinking levels exist between the different samples studied. A second concern is the selection of the crosslinking agent (60 Å-long). In principle this distance is not expected to significantly alter the conformational equilibrium, as it is similar, within experimental uncertainty with the highest distance between the fluorophore centers. A shorter crosslinker would be expected to have a more significant effect biasing the conformation of the SF, and be more efficient restraining dynamic states. Last but not least, the authors claim that the crosslinking reduces the flexibility of the SF, but without data to support this idea. If this claim is made, the authors should smFRET experiments in the crosslinked sample. While the rest of the manuscript adequately compares results in NaK and NaK2K, the crosslinking experiments are done exclusively with NaK2K. As a result, it is not possible to adequately compare the results with the conclusions of the rest of the manuscript. Performing crosslinking experiments with NaK would strengthen the manuscript. Taken together, significant new experiments, as described above, should be performed for the data to be able to support the claims made.

The electrophysiological data performed on the TREK2 channel was obtained in membranes of POPE:POPG (3:1). While this is consistent with the data for the other channel, this lipid composition is not physiologically relevant. This lipid mixture is often used as a simplified lipid composition that mimics that of the inner membrane of *E. coli*. However, it is far from the lipid composition found in a mammalian plasma membrane. The experiments should be repeated in a PC-based lipid composition.

Minor Comments:

The labeling of some figures should be revised:

- Figure 2 and 5. "Fret states" should be "FRET states", and "Lifetimes" should be "Lifetime"
- Figure 4 refers to the crosslinkers as "spacer", a term that the figure legend does not use. Crosslinker or Bis-MAL-dPEG is more accurate. Indicating the crosslinker concentration in the figure does not seem needed either.

Response to reviewer comments

Reviewer comments in black. Responses in red.

Reviewer #1

In this manuscript Matamoros et al. describe selectivity filter dynamics in the nonselective ion channel NaK, the K⁺ selective NaK mutant NaK2K, and the two pore eukaryotic channel TREK2. Following their prior work on KirBac1.1, the authors introduce cysteine mutants in the selectivity filter loop of both NaK and NaK2K. Fluorophores are attached to these sites and the single molecule FRET (smFRET) signal is measured in buffers containing 150 mM K⁺ or Na⁺. These smFRET measurements show that the selectivity filter of NaK2K is far more dynamic and less stable in a Na⁺ buffer compared to NaK. The smFRET data shows no ionic dependence. The authors then interrogate these channels in biological membranes by incorporation of L-Anap in COSm6 cells. These fluorescence measurements are consistent with the smFRET results. They then confirm these selectivity filter features are also observed in TREK2 using an extensive mutant (where native C residues are mutated to A) to allow for labeling specificity. This is a very nice study showing that selectivity filter dynamics and conformational states are important for most K⁺ channel classes, and not simply those undergoing C-type inactivation.

This is an extremely well written and well-organized manuscript. I find all the experiments convincing. The quality makes sense as this is the most recent in a series of FRET-based studies by the lab of Colin Nichols. My only criticism is that this study fits so well into this group's prior work that it does not feel entirely new. However, showing that sf dynamics extend beyond a prokaryotic Kir channel is important. I really like the new data showing that linkers restricting loop motions can abolish, to a degree, ionic selectivity. I think techniques of this type will be very important moving forward. I recommend publication in its current state.

We appreciate this very positive overall reception.

Reviewer #2

This report, entitled "Conformational plasticity of NaK2K and TREK2 potassium channel selectivity filters" is a thorough and well-designed study of the conformational dynamics of the potassium channel selectivity filter. The results presented in this work mirror a number of very recent structural studies of potassium channels, all of which converge on a theme, showing a dilation of the upper selectivity filter upon C-type inactivation or destabilization of the filter by removal of K⁺. The data in this study contribute to a growing convergent understanding of SF dynamics, though the observation that the non-selective NaK filter remains constricted is perhaps the most surprising result, as this data breaks with the overall trend showing dilation of the upper filter correlated with loss of K⁺ selectivity. The paper is clear and well written, the experimental techniques are appropriate for the questions being asked, and the results are intriguing, both for these specific channels and as an example of the use of this smFRET approach to explore SF dynamics in a range of ion channels. I have two major areas of identified weakness I hope the authors can address, and a few minor points that should be easily fixable.

Again, we appreciate this very positive overall reception.

1) The work by the authors showing that their findings in NaK2K are mirrored in the TREK2 channel are intriguing and consistent with crystallographic structures of TREK1 in high and low K⁺ (Manuscript Ref 25, Lolicato Sci Adv 2020). That being said, the S105 residue is a surprising choice for a probe of conformational changes within the selectivity filter. Whereas the T73 residue used for the smFRET studies of NaK2K is in a flexible loop region that borders the SF, TREK2 S105 is in the middle of the alpha helix that comprises TM1 and the helical cap above the membrane. S105 faces away from the selectivity filter in structural models of TREK2 and is approximately ~10-15 angstrom away from the nearest SF adjacent residue. While the choice of

this position may have been necessitated by the need to find a cysteine accessible residue for modification, it becomes hard to directly attribute changes in smFRET at this distant site to changes in selectivity filter structure, rather than being simply a readout of a more global conformational change in the protein.

a. To better understand the TREK2 results, I would ask the authors to perform an additional smFRET experiment with the TREK2 S105C construct in the presence of both 150 Na⁺ and the selectivity filter stabilizing drug ML335. If S105 does in fact report on the constricted versus dilated state of the filter, this drug should elucidate that. One would predict that ML335 should stabilize the SF in the conductive (and high FRET) state, counteracting the effect of 150mM Na⁺. This additional result would more closely link the S105 smFRET data to the structural state of the SF. We appreciate this issue and have made attempts to address the reviewer's very legitimate concerns. Regarding the choice of the labeling residue in TREK2, we agree with the reviewer's underlying point that the residue S105 is not the equivalent residue to T73 in NaK (which is in a flexible loop region at the borders of the SF). We continue to seek additional suitable positions, but this is a long and arduous task. The choice of the residue was affected by several factors. As the reviewer pointed OUT, the accessibility for cysteine labeling in the external part of the SF is limited due to the characteristic extracellular cap in K2P channels, and the number of residues accessible for modification is substantially reduced. We have tried to purify and label residue T182 which is ~equivalent to T73 in NaK, but the cysteine-less protein with the extra mutation (T182C) was not stable in our hands.

We have addressed the final comment with additional experiments, and we now show how ML335 shift the FRET histogram distributions to higher and more stable efficiencies in 150mM Na⁺, in good agreement with the prediction.

2) By using a PEG Maleimide crosslinker, the authors covalently trap a residue in the upper selectivity filter region in an extended state. Using single channel recordings, they demonstrate that the crosslinked NaK2K channels exhibit a decrease in single channel conductance for potassium and an increase in single channel conductance to sodium. I have several points to address in this section of the manuscript.

a. Data for these single channel recordings are somewhat incomplete. While changes in single channel conductance are reported, there is no discussion of other single channel parameters, most notably Po or open/closed channel dwell times. A more detailed analysis of these data would be helpful to understanding the results presented in this section.

b. The interpretation of the presented data states that the electrophysiological results are consistent with a change in ion selectivity for K⁺ versus Na⁺. However, the observed changes in conductance in experiments that utilize only mono-cationic solutions do not appear to conclusively support that assertion. The authors should more fully discuss their rationale for this claim. Alternatively, an additional experiment like the one presented in Supplemental Figure 2A, demonstrating a shift in reversal potential after reaction with the crosslinking reagent, would more definitely corroborate the claim of a shift in ion selectivity in the extended conformation after crosslinking.

We agree with the reviewer. We have now expanded the analysis of the single channel recordings, adding the Po, and have performed additional experiments as suggested. We show that there is a shift in the reversal potential and an increase in the Na⁺ current using KCl in the bath and NaCl in the pipette when we crosslink NaK2K with the 6 nm spacer (but not the 3 nm spacer), which indicates a decrease in the K⁺ selectivity in this condition.

c. From the methods, it appears that the PEG Maleimide was added to the bath and pipette solution during recordings, but the purified protein was not pre-incubated with the crosslinker. There is no description of the reaction rate of the PEG Maleimide with the channel or the time between application of the PEG Maleimide and initiation of the recordings. How do the authors

differentiate between channels that have been effectively crosslinked from those that haven't? The wide distribution of the individual data points in Figure 4c suggest a heterogeneous population of channels, perhaps consistent with incomplete modification of the T73C residue.

We have now reworded and added the details. The reviewer is correct; we incubated the liposomes with the crosslinker in the bath for > 5 minutes prior to sealing, and then, after patch excision we incubated another 5 minutes, before starting recording. We agree with the reviewer, that we most likely have a heterogeneous population of crosslinked, partially crosslinked and non crosslinked channels. Unfortunately, we cannot experimentally differentiate between these populations, hence we provide a large number of experiments per condition, to capture the ensemble.

Minor Points –

1) Page 5, line 118 – The manuscript reads “If the trace data are very noisy, cross-correlation analysis between donor and acceptor fluorescence intensities can be performed.” The language here is a bit confusing. By noisy, do the authors mean instances where the FRET signal is dynamic?

We agree with the reviewer that the sentence is confusing, and the answer is yes, the signal is more dynamic. We have modified the sentence to make it more specific and clearer.

2) Supplemental Figure 2c, A53An mutant. While not central to manuscript, I am confused by the results for this mutant in NMDG. With Na⁺ eliminated as the cation responsible for depolarization, this mutant still is significantly depolarized relative to untransfected cells. Is the assumption that this mutant also allows NMDG to permeate through the NaK2K channel?

This is an astute observation, and the answer is yes, the experimental data suggest that this particular mutant allows NMDG to permeate through the NaK2K channel, and potentially could explain why there is no shift in the A53Anap spectra.

3) Figure 3, panel A – Identifying the locations of the introduced mutations is somewhat difficult. I would suggest enlarging the structural models and revising the color coding in the models, as they do not appear to match the provided key.

We agree, the figure has been modified to make it clearer.

4) The authors might also like to cite a recently published study on inactivated conformations of the Kv1.3 channel, with similarities to the presented results: Selvakumar, P., Fernández-Mariño, A.I., Khanra, N. et al. Structures of the T cell potassium channel Kv1.3 with immunoglobulin modulators. Nat Commun 13, 3854

Thank you for the suggestion. Reference added.

Reviewer #3

The manuscript by Matamoros et al. studies the conformation of the selectivity filter (SF) of two potassium channels, the bacterial NaK and the mammalian TREK2. Two versions of NaK are used, the WT and Na2K2 (obtained by mutation of the SF), which is selective for K⁺ over Na⁺. This is a high-quality paper that uses interesting complementary approaches, including activity assays. However, I have reservations with regards to the impact/significance, and with some experimental components. The same laboratory previously (Refs 21 and 22 of the manuscript) studied by single-molecule (sm) FRET the SF of another type of K⁺ channel (KirBac1.1). These studies showed changes in the SF filter, which becomes more dynamic in the presence of Na⁺. The current work is somewhat confirmatory, as it shows similar observations for NaK2K and TREK2. This is still relevant because it suggests that this dynamic change could be a general behavior across a wide range of K⁺ channels. However, the current data it is not particularly

ground-breaking, in particular because a different group performed ssNMR of NaK2K, providing data that showed the same general conclusions. These circumstances significantly reduce my enthusiasm for this otherwise strong and carefully done work.

We appreciate the positive overall reception.

The crosslinking experiments are the weak element of the manuscript. I have serious reservations that the use of the crosslinker is indeed providing the information that the manuscript claims. First of all, the crosslinking scheme allows for different types of crosslinking events; due to the presence of four Cys, a crosslinking pair could have a distance of 28 Å, or 35 Å. As a result, one can expect that the crosslinked samples contain an unknown mixture of crosslinked populations. More worrisome, more than one crosslinker could be engaged in a single channel. In these circumstances, a careful examination of the product of the crosslinking should be performed. However, the manuscript does not contain these studies. A simple SDS-PAGE should be able to 1) identify the presence of crosslinking, and 2) characterize the resulting samples (more advanced techniques would be welcome). Importantly, comparison between different experimental conditions (Na⁺ vs K⁺) would identify if comparable crosslinking levels exist between the different samples studied.

We agree with the reviewer, the crosslinked samples contain an unknown mixture of crosslinked populations and yes, more than one crosslinker can engage in a single channel. Unfortunately, we cannot differentiate between these populations, requiring us to perform a good number of experiments per condition, in order to 'capture' the ensemble. We have addressed the concern about demonstrating the presence of crosslinking by western blot experiments with antibodies against the C-terminal poly histidine tail of NaK and NaK2K. These indeed show increased tetramer band in the presence of crosslinking agent. The experiments are not simple and straightforward, due to the fact that NaK2K in particular is intrinsically stable in the tetramer configuration in SDS-PAGE. To counter this, we destabilized the NaK2K tetramer, after crosslinking, by treatment with high concentration of guanidinium hydrochloride for 16 hours, before running the SDS-PAGE. Even this does not completely monomerize the wild type protein, but does provide evidence that tetramers are specifically increased by the crosslinker presence for both NaK and NaK2K.

A second concern is the selection of the crosslinking agent (60 Å-long). In principle this distance is not expected to significantly alter the conformational equilibrium, as it is similar, within experimental uncertainty with the highest distance between the fluorophore centers. A shorter crosslinker would be expected to have a more significant effect biasing the conformation of the SF, and be more efficient restraining dynamic states.

We apologize for the wording that could led to a mis-implication. Here, we selected the crosslink agent based on the low FRET state diagonal distances (60 Å), having a high FRET diagonal distances of 40 Å. Our hypothesis was that NaK2K would behave different if we could dilate the SF from the original high fret state (predominant in high K⁺) to the lower FRET states levels (predominant in high Na⁺). We have performed additional multi-channel recording experiments with a 3 nm crosslinker. While western blotting provides evidence that crosslinking occurs, we do not see any statistically significant differences between control and crosslinked samples in current properties, either in NaK or in NaK2K.

Last but not least, the authors claim that the crosslinking reduces the flexibility of the SF, but without data to support this idea. If this claim is made, the authors should smFRET experiments in the crosslinked sample. While the rest of the manuscript adequately compares results in NaK and NaK2K, the crosslinking experiments are done exclusively with NaK2K. As a result, it is not possible to adequately compare the results with the conclusions of the rest of the manuscript. Performing crosslinking experiments with NaK would strengthen the manuscript. Taken together,

significant new experiments, as described above, should be performed for the data to be able to support the claims made.

We agree with the reviewer, and have now reworded the sentence trying to be more cautious in our claims. Unfortunately the experiments that the reviewer is suggesting - applying smFRET to the crosslinked sample - are technically not possible at this time. However, as the reviewer suggested, we have performed single channel recording experiments with NaK and now include these in the manuscript. We do not see any statistically significant differences between control and crosslinked samples in Na⁺ or in K⁺. This provides further consistency with the overall hypothesis that K selectivity in NaK2K requires SF flexibility to adopt the K-selective conformation, but that non-selectivity in NaK does not.

The electrophysiological data performed on the TREK2 channel was obtained in membranes of POPE: POPG (3:1). While this is consistent with the data for the other channel, this lipid composition is not physiologically relevant. This lipid mixture is often used as a simplified lipid composition that mimics that of the inner membrane of E. coli. However, it is far from the lipid composition found in a mammalian plasma membrane. The experiments should be repeated in a PC-based lipid composition.

Poor original wording here led to a mis-implication. The electrophysiological TREK2 data were obtained in soybean lipids, which are 47.5% PC.

Minor Comments:

The labeling of some figures should be revised:

-Figure 2 and 5. "Fret states" should be "FRET states", and "Lifetimes" should be "Lifetime"
Corrected.

-Figure 4 refers to the crosslinkers as "spacer", a term that the figure legend does not use. Crosslinker or Bis-MAL-dPEG is more accurate. Indicating the crosslinker concentration in the figure does not seem needed either.

Corrected. We have removed the term 'spacer' throughout.

Reviewers' Comments:

Reviewer #2:

Remarks to the Author:

The authors have now addressed my concerns with the original manuscript.

The result showing that ML335 shifts the TREK2 FRET signal to a high-FRET state when TREK2 is in 150 Na⁺ suggest that S105 does serve as a reasonable reporter on the state of the selectivity filter. While the PEG-Maleimide crosslinking data remain the least methodologically definitive section of the manuscript (as the authors acknowledge in both their response to reviewers and in the manuscript), these results still contribute to the overall message of the paper. The data are consistent with other recent reports of ion dependent selectivity filter dynamics and will be an important contribution to the field.

Reviewer #3:

Remarks to the Author:

The additional experimental performed, plus the rewording of the manuscript satisfy me. I have no further objections to this work.

Reviewer #4:

Remarks to the Author:

The manuscript entitled "Conformational plasticity of NaK2K and TREK2 potassium channel selectivity filters" written by Matamoros et al. is a well-written work with interesting complementary approaches. I appreciate the well-designed study, careful interpretation of results, and the engagement of multiple systems to address the findings that the dynamics of the selectivity filter are crucial for the ion selectivity of many K⁺ channels. I only have a few minor points towards some technical details for clarification.

- 1) The authors described that proteins without intrinsic cysteine were used as labelling controls to examine unspecific labelling. What is the common percentage of protein being nonspecifically labelled? How was it compared to a sample with cysteines introduced, and was labelling efficiency routinely examined for each sample? A fluorescence gel with samples with or without cysteines that underwent the same labelling procedures would be helpful for comparison. Other means that can address the same issue are welcome.
- 2) How was a protein:lipid ratio of 1:200 determined? What is a typical reconstitution efficiency for the samples, and do the authors believe at this ratio, a single molecule level was achieved? If this follows a Poisson distribution, what would be the estimated percentage of liposomes that contain more than 1 channel, and were there other measures to exclude those molecules?
- 3) Were the independently labelled samples from different protein purifications or transfections?
- 4) Figure 1 panel C, representative traces shown are all over 45 seconds. However, contour plots and histograms are only using data from the first 3 seconds. Were most of the traces long, or at 3 seconds, they already suffer significantly from the stochastic photobleaching events? would the results change if the author presented 10-second or at least 5-second long data?
- 5) Figures 2 and 5. I have to point out that this idealization is not one of the best. Some of the raw states are clearly outside the idealized black line. Typically, the three Gaussians for NaK2K under a NaCl condition don't seem to fit well. Can the authors comment more on this, the model used for idealization, and would the authors consider potential sub-states?

Response to Reviewer #4: Response in red

The manuscript entitled “Conformational plasticity of NaK2K and TREK2 potassium channel selectivity filters” written by Matamoros et al. is a well-written work with interesting complementary approaches. I appreciate the well-designed study, careful interpretation of results, and the engagement of multiple systems to address the findings that the dynamics of the selectivity filter are crucial for the ion selectivity of many K⁺ channels. I only have a few minor points towards some technical details for clarification.

1) The authors described that proteins without intrinsic cysteine were used as labelling controls to examine unspecific labelling. What is the common percentage of protein being nonspecifically labelled? How was it compared to a sample with cysteines introduced, and was labelling efficiency routinely examined for each sample? A fluorescence gel with samples with or without cysteines that underwent the same labelling procedures would be helpful for comparison. Other means that can address the same issue are welcome.

The non-specific labeling is typically <10% of the labeling observed in the cysteine mutants, and essentially never shows fluorescence traces that fulfil the criteria for single molecule FRET (i.e. single step photobleaching of both fluorophores, correlated fluorescence between fluorophores) that would provide false positive FRET traces. Previously we have shown that there is essentially no fluorescent labeling using the same approach for cysteine-less K channels incorporated into liposomes (inset Figure), and we have not systematically analyzed labeling efficiency here.

REDACTED

2) How was a protein:lipid ratio of 1:200 determined? What is a typical reconstitution efficiency for the samples, and do the authors believe at this ratio, a single molecule level was achieved? If this follows a Poisson distribution, what would be the estimated percentage of liposomes that contain more than 1 channel, and were there other measures to exclude those molecules?

This ratio was arrived at in previous studies on KirBac1.1 to optimize the number of single molecule FRET traces and minimize the number of multi-molecular traces. Assuming Poisson distribution with $\lambda=1$, there should be ~37% of vesicles with 0 channels, 37% with 1 channel, and 26% with more than one channel. However, for

The specificity of fluorophore labeling in KirBac1.1 cysteine mutants examined by SDS-PAGE. The labeled protein samples were separated by 4-20% SDS-PAGE, and fluorescence images were acquired with a Kodak fluorescence scanner. Coomassie staining shows similar levels of protein for WT and Cys mutant proteins, but Alexa Fluor 555 emission (600nm) and Alexa Fluor 647 emission (670m,) are both largely absent from ‘labeled’ WT protein. [From Wang et al., 2016, *Nature Structural & Molecular Biology* 23: 31–36, Supp. Fig. 1C].

Colin G. Nichols FRS, `Carl Cori Professor and Director of CIMED

Washington University School of Medicine at Washington University Medical Center, Campus Box 8228, 660 S. Euclid Avenue, St. Louis, Missouri 63110-1093 (314) 362-6630 Fax: (314) 362-7463
cnichols@wustl.edu

smFRET analysis, the criteria that there be single step bleaching of both donor and acceptor fluorescence ensures that only single donor- and acceptor-labeled channels are analyzed.

3) Were the independently labelled samples from different protein purifications or transfections?
Independently labelled samples were generally from different protein purifications.

4) Figure 1 panel C, representative traces shown are all over 45 seconds. However, contour plots and histograms are only using data from the first 3 seconds. Were most of the traces long, or at 3 seconds, they already suffer significantly from the stochastic photobleaching events? would the results change if the author presented 10-second or at least 5-second long data?

Most of the traces were long, but photobleaching becomes problematic after more than a few seconds, and only traces >3 seconds were analyzed. A fixed time of three seconds was chosen to maximize the number of individual traces with time-stable behavior (evidenced by the time-stable averages), and to avoid over-representation of individual long traces.

5) Figures 2 and 5. I have to point out that this idealization is not one of the best. Some of the raw states are clearly outside the idealized black line. Typically, the three Gaussians for NaK2K under a NaCl condition don't seem to fit well. Can the authors comment more on this, the model used for idealization, and would the authors consider potential sub-states?

We agree that the three Gaussians are not a perfect fit, and that the idealization could be made with more states. However, as we now show by overlapping the actual FRET histograms and the idealized state distributions, three states does capture the most prominent components. As we have pointed out in the text, this is inevitably an oversimplification.